# The Ripple Framework: Co-Producing Social, Cultural, and Economic Value in Care Through a Generative and Relational Approach

**DOI:** 10.3390/ijerph21111521

**Published:** 2024-11-15

**Authors:** Luis Soares, Sarah Kettley

**Affiliations:** The Institute for Design Informatics, Edinburgh College of Arts, College of Art, Humanities and Social Sciences, The University of Edinburgh, Edinburgh EH3 9DF, UK; sarah.kettley@ed.ac.uk

**Keywords:** design research methods, radical participatory design, relational research, collaboration, complexity, ripples, value, trust

## Abstract

Work has been undertaken in the healthcare sector to explore ways of co-producing design responses with different communities and organisations. However, we lack empirical analysis of how design thinking can help tackle complexity. To assist the Healthier Working Lives programme, we curated the Ripple Framework to develop trust and attempt to address the challenges of complexity in residential care. Through a generative design process, data were used to collaboratively define bespoke co-design pathways with 31 participants from six Scottish care providers over ten months. Thematic, content, and matrix analyses produced insights to inform vignettes illustrating how design responded to complex social care sector needs, with a particular focus on the fulfilment and flourishing of the care workforce. Drawing on our empirical material and using the Design Research Value Model, we illustrate how we have developed social, cultural, and economic value in care through co-design, enabling an opportunity to test the novel methodology.

## 1. Introduction

New conceptual lenses to magnify micro interactions are needed to assist us in unveiling the complexity of everyday practices. However, this might not suffice since new forms of engagement are also required. This article attempts to steer design research processes towards a space for social interaction and exchange. We are conscious that this not only requires perseverance, but also calls for collaboration to help translate meanings entangled in the panoply of languages and experiences that make up everydayness [1]. Scharmer [2] appeals for more awareness and self-consciousness to allow individuals and groups to shift from their habitual places to embrace the future, whether it be preposterous, possible, plausible, projected, probable, or preferable [3]. These two dialogical perspectives (i.e., awareness and self-consciousness), conceptual and applied, suggest that when people adopt a prospective approach and begin to operate from a future space of possibility, they are embracing a new lifeworld attitude (presencing) which encompasses new “meanings and shared languages that allow individuals to empathically interact with one another [and experiment] new forms of figuration and meaning” [4]. Drawing on [5], we want to conceptualise from within a context of experiential research, placing the individual experience at the centre of inquiry. This should allow us to observe but also to purposefully inhabit, embody, and explore a particular context, problem, or question. We expect this will enable engagement with dimensions of human experience, which can be harder to access through traditional third-person methods. Essentially, this kind of research involves everyone taking part to reflect on their own experiences, thoughts, feelings, and actions [6] in relation to the problems or questions they are investigating. It can help researchers to see ‘inside’ the experience of the users, to better understand and meet their needs. It can also lead to new insights and innovations that might not be apparent from a more detached perspective. This paper set out to identify ways to promote healthier working lives and ageing for older care workers in Scotland. The paper contributes two perspectives to the field of Design and Healthcare scholarship: (1) a novel methodological approach, the Ripple Framework, developed to help improve new forms of social interaction between designers and co-designer communities based on trust and non-judgemental relationships, and (2) an illustration of how valuation (Social, Cultural, and Economic) can be derived from a more designerly way of knowing [7] which can be used to inform policy making [8].

### Context

According to a recent study, “approximately 13% of the total UK workforce is employed in the health and care sector” [9]. Amongst a set of complex and interconnected factors, a lack of training and professionalization [10,11,12] emerges as a key factor contributing to the demoralization of the care workforce, which is exacerbated by a lack of recognition and rewards, as stated by [13], therefore increasing the rate of turnover. The aftermath of the COVID-19 pandemic exposed the lack of preparedness and resilience of the care sector, not only from the human resource point of view [14] but from an infrastructural perspective [15]. This state of affairs can be described as a “wicked problem” [16], which calls for new approaches [17] to help address complexity and uncertainty [18,19], which underscores the impetus for this research.

## 2. Materials and Methods

To illustrate how the social, cultural, and economic values of care can be derived through a relational approach, this article is developed using co-produced design outcomes from the Healthier Working Lives (HWL)) project, a research initiative funded as part of the UKRI funded Health Ageing Challenge Social, Behavioural and Design Research in the UK, an interdisciplinary, intersectorial programme, which attempts to find ways of improving working conditions for care workers. Co-design was used to develop engagement, aiming to bring to the surface relevant aspects of the care workforce experience and culture and to enable the co-production of alternatives to facilitate the carrying out of day-to-day activities. Nevertheless, numerous challenges had to be overcome to successfully conduct this research, the most significant of which was the fact that this study took place during the UK COVID-19 lockdowns, which introduced a high degree of uncertainty and distrust. To address this, a methodological tool called the Ripple Framework was developed. This tool provided the flexibility to dynamically redesign co-design pathways in response to any unanticipated situations encountered by the research teams. This strategy was instrumental in building and reinforcing trust. Ethnographic data from 44 interviews enabled us to approach each care home as a unique case, informing the definition and selection of tailored co-design activities for them. The co-design process was implemented in five stages for each home, with each stage having a specific action and intended outcome (see Table 1), ultimately establishing a customized pathway for each care home (see Figure 1, Figure 2, Figure 3, Figure 4, Figure 5 and Figure 6).

An ethnographic pre-design phase involved an open-ended and exploratory empirical engagement with day- and night-shift workers aged 50 and over, providing a profound understanding of the frontline workers’ experiences [20]. (The pre-design, or ethnographic, phase involved open-ended exploration with day- and night-shift workers aged 45 and above. This focus aligns with The Adult Social Care Workforce in Scotland report, which states that in 2020, eight out of ten (80%) adult social care staff in Scotland were female, and 44% of workers were aged 45 and over. These workers averaged 31 h per week, with the majority (86%) employed on permanent contracts. The same report indicates that the adult social care sector was predominantly white, with at least 69% of the staff identifying as such. Interestingly, there appears to be a higher proportion of ethnic minority staff in the private sector compared to other sectors. Overall, 3% of the workforce reported belonging to an ethnic minority, ranging from 2% in the public sector to 5% in the private sector. (see: https://www.gov.scot/binaries/content/documents/govscot/publications/research-and-analysis/2022/06/national-care-service-adult-social-care-workforce-scotland/documents/adult-social-care-workforce-scotland/adult-social-care-workforce-scotland/govscot%3Adocument/adult-social-care-workforce-scotland.pdf, accessed on 10 October 2024). Building on these insights, each session in every care home was tailored according to the findings of the preceding activities (i.e., it was a generative design research process). A diverse array of methods was employed, including walkarounds, probes, performative narrative interviews, experience-based co-design, card sorting, table games, serious play, and crafting postcards and newspaper headlines envisioning the future (see https://ripple.designinformatics.org/, accessed on 10 October 2024). Each activity was strategically designed to enable trust to flow.

After the initial walkaround phase, where we posed broader questions about the roles and experiences of care workers, we moved on to stage two. We utilized poster probes to present four existing healthcare projects and technologies, prompting questions such as, “How could technologies or initiatives like these support your work?” In the third stage, we employed Performative Narrative Interviews, which featured more open-ended questions, including, “What does teamwork mean to you, and how important is it in delivering care?”, “How does a lack of recognition affect care?”, “What does self-care mean to you?”, and “What barriers prevent you from delivering care?” These questions were developed based on insights gathered from follow-up interviews with each care home after the poster probes were collected. By using semi-structured approaches, we aimed to create a sequence that would deepen and establish meaningful relationships with the 31 participants who engaged with us over the ten months of co-design. To achieve this, a strategic approach was devised, involving collaboration with The Scottish Care, a research partner closely connected to Scottish care homes. This organisation served as a gatekeeper, assisting in the selection of care homes for the research. The selection process carefully balanced criteria such as rural and urban locations, as well as varying business models (e.g., family-owned vs. private). This approach proved to be an effective sampling strategy, as the trust established between the care homes and that organisation facilitated smooth and efficient access.

### 2.1. Data Collection and Analysis

Data were collected through interviews and co-designed artifacts—such as cards from the card sorting activity, crafted postcards, and mock front pages of future newspapers—alongside verbatim transcriptions of audio recordings from the co-design sessions. To analyse these data, we employed thematic analysis, content analysis, and matrix analysis, which helped to uncover the meanings and shared values expressed by our co-designers. Building on these findings, we conducted a deliberative session (stage four) with each of the six care homes. Through open debate, we explored the meanings, emotions, and expectations conveyed through language, working towards a prioritized list with each care home team of the issues that staff identified as needing change. The Futuring activities in stage five allowed participants to project their expectations and envision scenarios involving these priorities, playing across preposterous, possible, plausible, projected, probable, and preferable futures [3]. We emphasize that the bespoke design and delivery of these six co-design pathways (Figure 1, Figure 2, Figure 3, Figure 4, Figure 5 and Figure 6) was made possible by utilising the Ripple Framework, which was specifically curated for this purpose, and which is presented next.

**Figure 1 ijerph-21-01521-f001:**
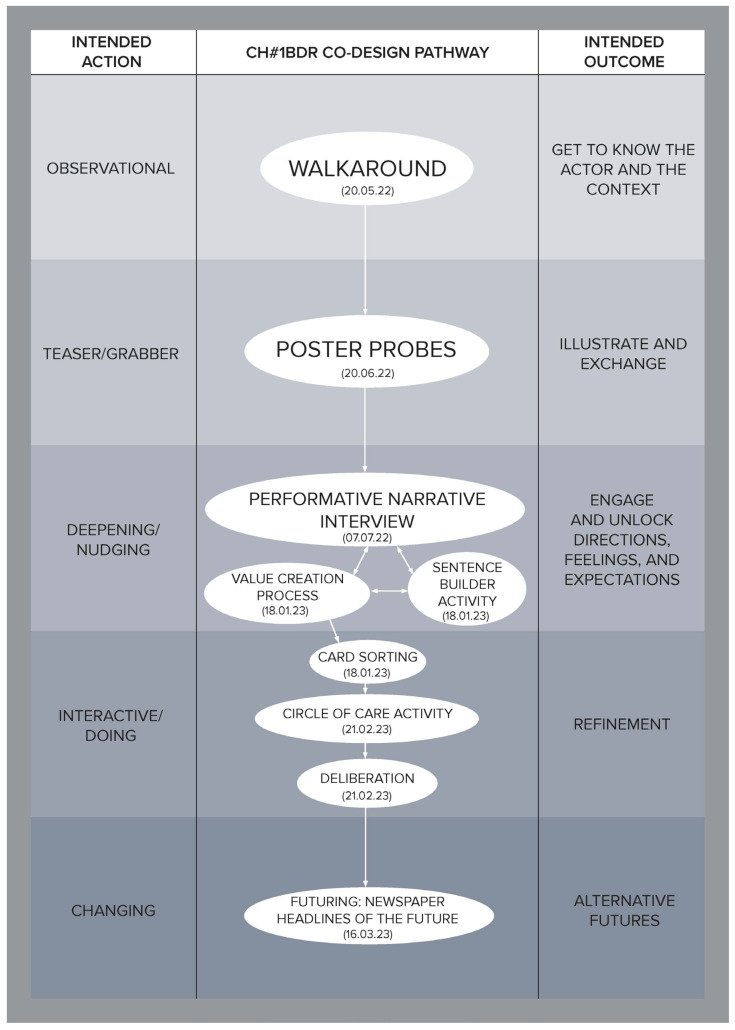
Co-design pathway for CH#1.

**Figure 2 ijerph-21-01521-f002:**
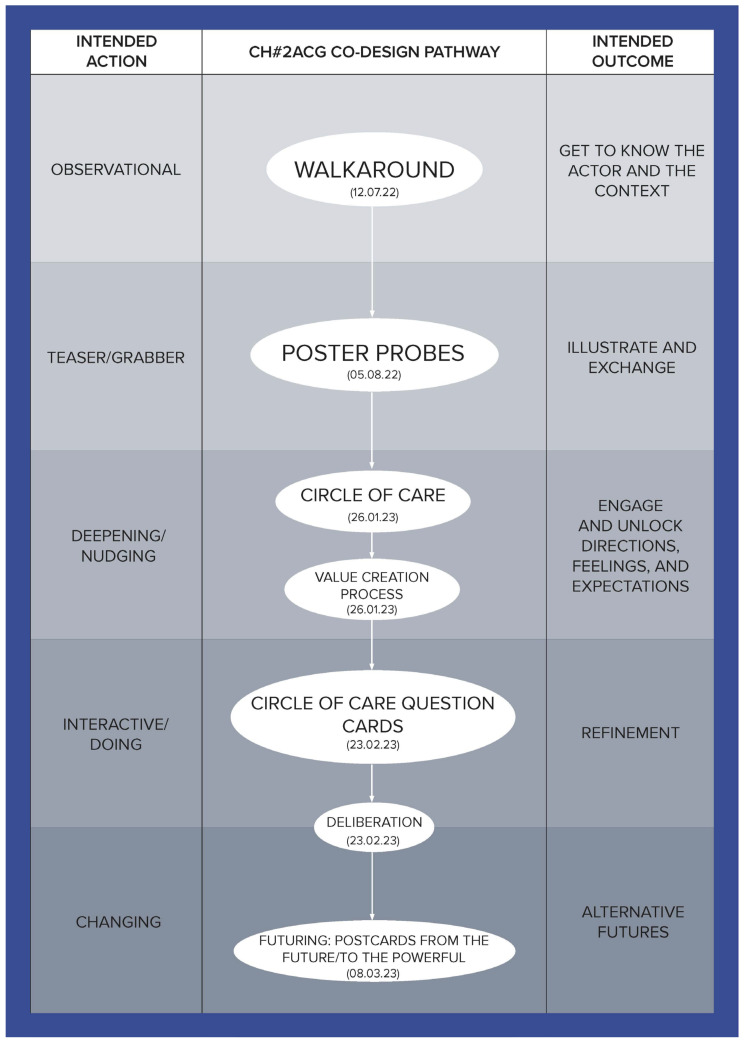
Co-design pathway for CH#2.

**Figure 3 ijerph-21-01521-f003:**
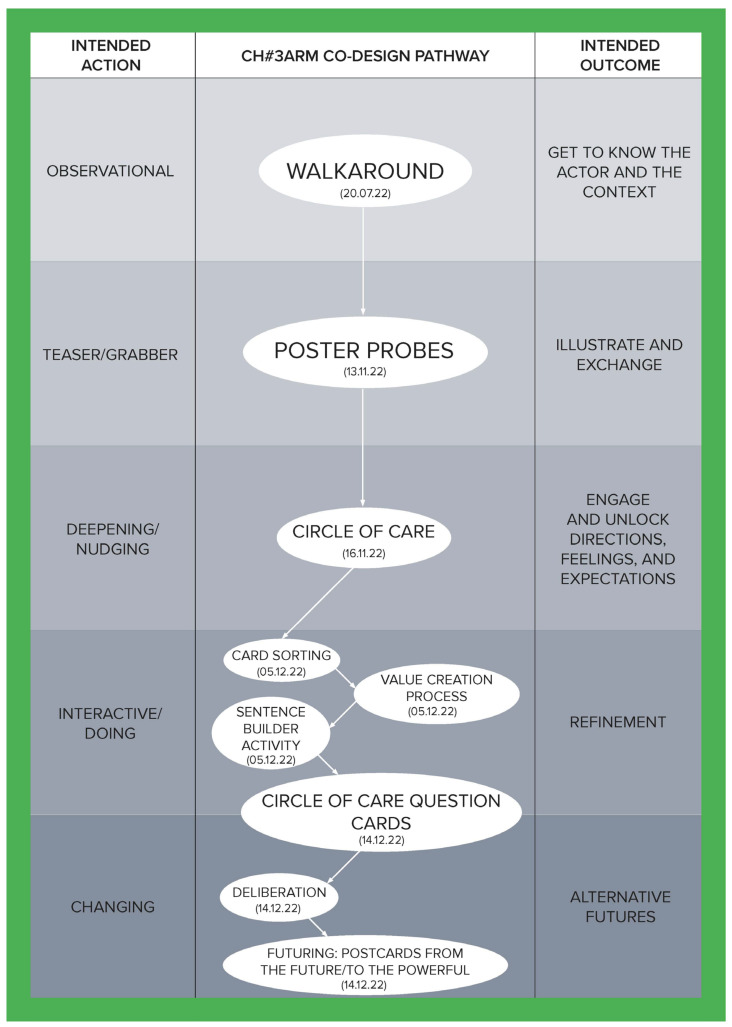
Co-design pathway for CH#3.

**Figure 4 ijerph-21-01521-f004:**
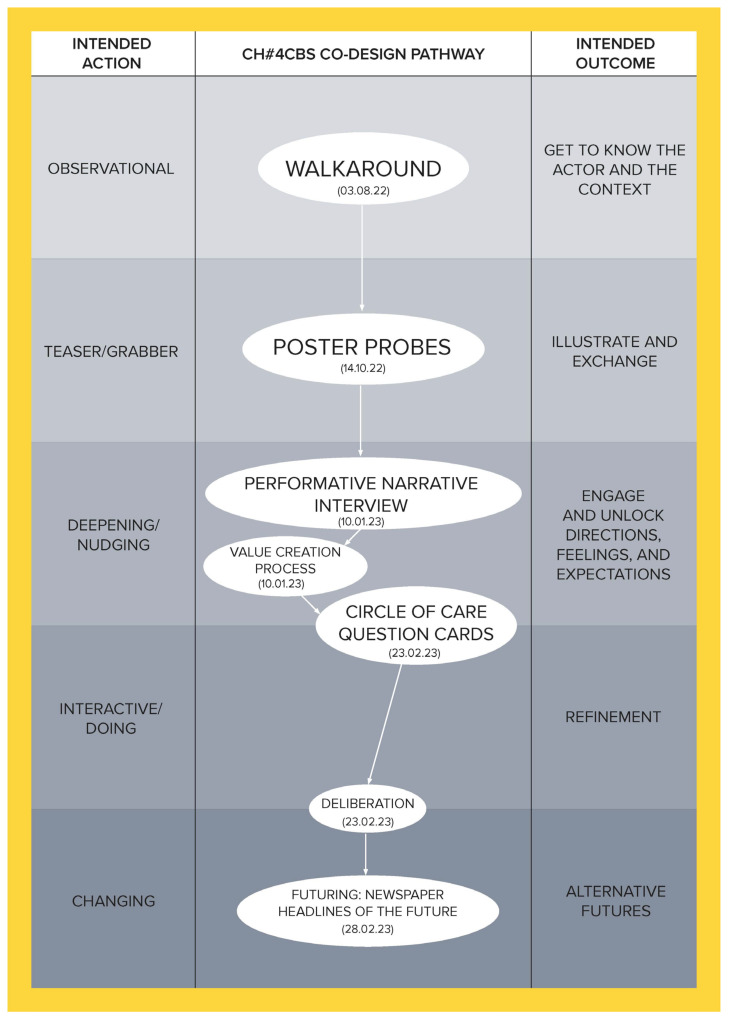
Co-design pathway for CH#4.

**Figure 5 ijerph-21-01521-f005:**
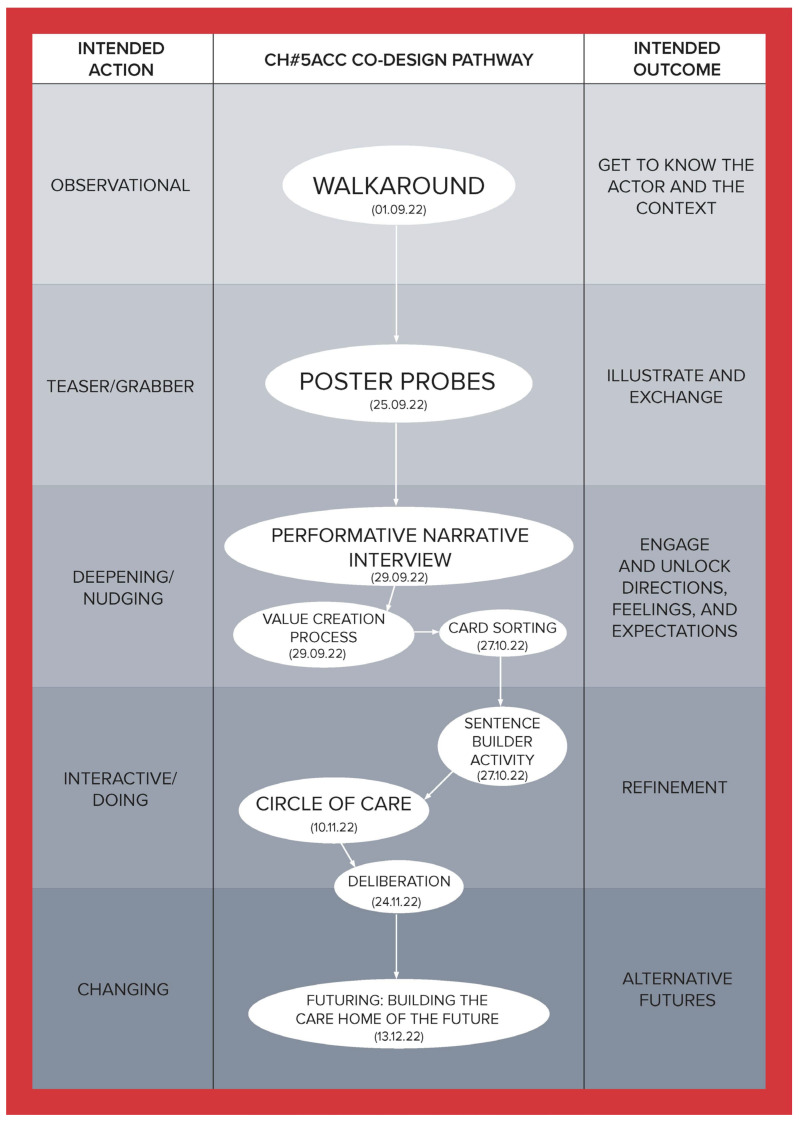
Co-design pathway for CH#5.

**Figure 6 ijerph-21-01521-f006:**
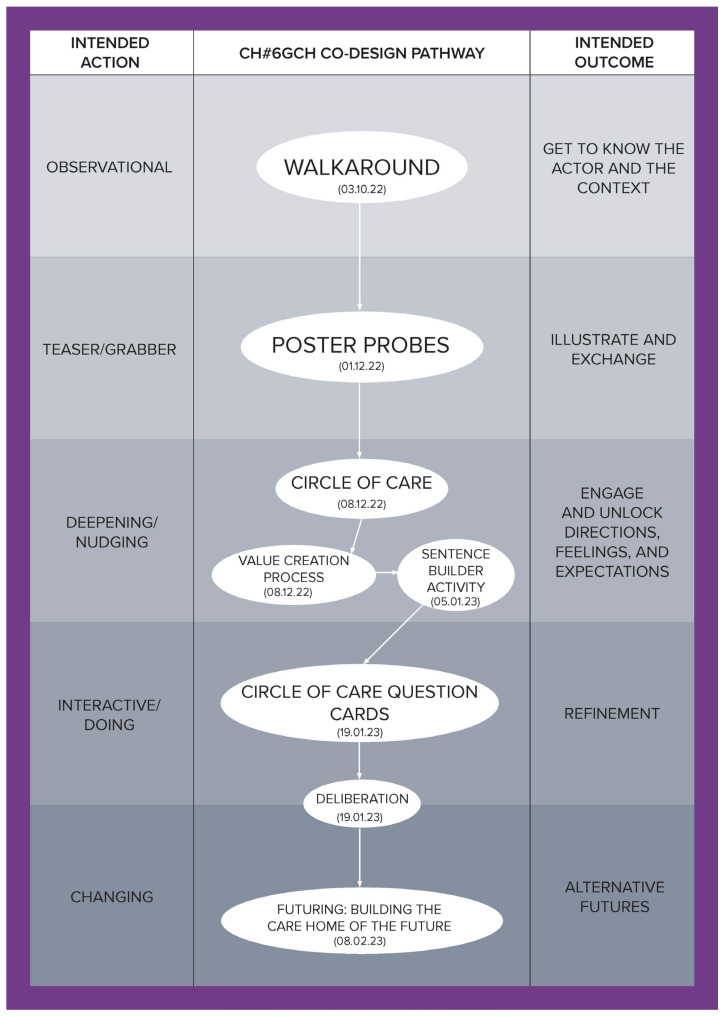
Co-design pathway for CH#6.

### 2.2. The Ripple Framework

Vines et al. suggest considering “peoples’ current practices, experiences…and how can we make best use of these [to co-produce design responses]” [21]. Within this frame of reference, we developed the Ripple Framework (RF) (Figure 7), a methodological framework to help promote a more “designerly” perspective [22] to design research while being responsive to unpredictable conditions in the field. Debrah [23], building on [24], emphasize that design practice “[…] in certain contexts is devoid of user inputs”; consequently, when proposing the Ripple Framework as a methodological tool, we aim to promote the radical participatory design by placing the target audience at the centre of the process, allowing them the opportunity to reclaim their agency to think, talk, and act in a more relational manner. The framework will enable more flexibility to tackle complexity.

## 3. Background Work

Building on the foundational work of Udoewa [25,26,27], this article advocates for a more relational approach to research. This perspective challenges the traditional reliance on empathy as a justification for design decisions, proposing instead an ontological shift that emphasizes “being alongside the other” [25]. By adopting this stance, the approach acknowledges and respects the situated design agency of individuals, rather than assuming their role or position. In this way, we see the development of the Ripple Framework not only as a pragmatic tool but as a platform for developing partnerships to animate collective futures [28]. By seeking to familiarise the care workforce with co-design methods and frameworks, we intend to generate not one-off solutions to problems, although these may also be valued outcomes according to Knutz and Markussen [29], but a conceptual space for ongoing creative autonomy and power. This process involves the staff actively redesigning their own work conditions, using commonly available resources as the foundation for developing new mechanisms to enhance health and well-being. This approach is necessarily grounded in a relational ontology, moving away from methodological individualism to focus on the emergent relationships between roles and individuals, and the interactions between people and their environments.

Applying this to the HWL, we argue that radical participatory forms of participation [26,27] favour increased levels of creativity in psychosocial care. Therefore, this work aims to contribute to the well-being needs of the care workforce through a relational approach to co-produce design responses, which goes beyond transactional requirements to gather information; rather, information is understood as something that is creatively generated [30]. A human-centred perspective is essential to this form of relational design. This approach allows researchers and co-researchers to fully immerse themselves in the lifeworld, making it a critical component of the research process. This can lead to a deeper and more nuanced understanding of the communities’ needs and preferences that can inform design decisions. It also allows researchers to test and refine their designs in real-world contexts, addressing problems and obstacles as they arise. Moreover, by centring the research process on relations or relationships, we argue that it can produce knowledge that is not only theoretically informed but also personally meaningful and actionable. We believe this approach will facilitate the activity of describing, explaining, and ultimately generating ideas and actions adequate to the context of action and actors that inhabit those complex lifeworlds.

This work also aligns with Bate, who argues in favour of a new approach to capture the “messy, debatable and unquantifiable but essentially human dimensions of life, such as history, beauty, imagination, faith, truth, goodness, justice and freedom” [31]. Here we make a point by stressing that design research has better chances of success if performed in the “wider context of human interaction with the world” [32]. We see this as extending co-design activities considering that we do not perceive the design process as “distinct from ‘being’ or the ‘ongoing flow’ of daily life, and from the dynamic complexity of the lifeworlds of users” [4].

### 3.1. Everyday Life and Complexity

Social life is complex; it is an undeniable truth, whether one likes it or not. However, complexity does not need to be understood as “evil” [22]. For instance, both in physics and information theory, this phenomenon is referred to as entropy and can be tracked back to Shannon’s information entropy theory [33] which states that “the entropy of the output can be calculated” to try and give a degree of coherence to complex phenomena [34] in a form of more desirable assemblage.

Traditional methods for addressing these challenges have often relied on a positivistic perspective, employing mechanistic and linear reductionist techniques (as per Figure 8) that are more appropriate for physical systems rather than complex adaptive human systems. As a result, they have not succeeded in achieving the required system transformation [35]. For instance, reductionism seeks to simplify complexity by reducing it to its constituent parts. However, while this approach can yield insightful results, it often overlooks critical facets of reality that cannot be easily reduced to simpler elements. These overlooked aspects often entail the convoluted dynamics (noise and uncertainty per Figure 8) that configure a given lifeworld or the lived world of social reality. However, our lifeworlds are constructed through different discourses (patterns as per Figure 8), performed in different languages, embedded in different cultural matrices, and enacted through different social practices and bodies. Each of these spheres carries its unique complexity that cannot be easily reduced (insights as per Figure 8) to simpler elements without losing critical aspects of their reality.

In other words, the simple clarity and focus suggested in Figure 8 is often not so simple, and the complex cannot be easily discarded as merely complicated. Moreover, each of these spheres of everyday discourses often generate different forms of dissensus or disagreement rather than consensus. This dissensus is not merely a matter of different views or perspectives, rather it is a manifestation of the inherent tensions, conflicts, contradictions, and contestations within and between different spheres of life. Therefore, while reductionism and simplicity may offer appealing avenues for investigation and understanding, they may also lead to misconceptions, misinterpretations, or oversimplifications. It becomes crucial then to take this into consideration in the design thinking process, namely when moving from research and synthesis to concept and prototyping, and from there to design response (see Figure 8). This is an important matter to be addressed to assist the design community in navigating the “messy and wicked [character of] everyday context” [22].

### 3.2. Addressing the Well-Being of the Care Workforce

Design thinking, specifically in health care, typically places an emphasis on the relationship between care providers and patients, often focusing on barriers and opportunities for the deployment of technologies for the elderly recipients of care [36]. This highlights the need to better model [5] and integrate technological resources into the day-to-day care practice [37] and to provide social and emotional enrichment [38] to both care workers and residents. In harmony with this body of literature, and those arguing in favour of holistic approaches [39], we stress the need to refocus designed alternatives based on a balanced trade-off between care providers (in this case the care workers) and beneficiaries (residents) [40].

Wearable technology development has seen some healthcare research, addressing the psychosocial needs of technology wearers as bodies in relation with the materials and forms of technology and with viewers [41]. Service design, with its emphasis on stakeholder maps and power relations between roles, has highlighted a sequence of relations that is in many ways relational [42,43,44]. Therefore, when thinking about adequate responses, the need to consider the different forms of values (social, cultural, and economic; see Figure 9 and day-to-day priorities of residential care staff as users emerges.

Consequently, “[u]nderstanding what capabilities matter and how to cultivate them is especially important in care settings because of the complex, unstructured, dynamic and unpredictable nature of these locations” [46]. We find complementary research on the relational in other disciplines, notably well developed in psychotherapy [47,48], and in ontological and new material approaches in craft theory and the materially led design of technology [35,49]; Signals of a similar ontological shift to the relational (in the west) can also be seen in the public health literature [50,51].

## 4. Results

By adopting a relational and dialogical approach to co-producing change, the HWL introduced a new design environment where the health, dignity, and well-being of staff are inextricably linked to the systemic issues they face. To achieve this, we tested an activity called the Circle of Care (CoC), which goal was to support strategies aiming at flattening vertical hierarchical structures. The activity began with recognising the true value of care work, improving the work environment, elevating the status of caregivers, offering personal development opportunities, and ensuring that caregivers themselves are well cared for. Key issues that need to be addressed include the following:Providing basic physical aids for routine tasks;Enhancing mental health support;Improving communication tools;Implementing measures to elevate worker recognition and status in society.

Some responses to these needs became apparent through the use of the CoC and other methods included in the Ripple Framework, as will be demonstrated using the vignettes technique. This approach lays the foundation for understanding how social, cultural, and economic value can be generated through design research.

### 4.1. Vignettes

To revisit some of our stories, we employed the technique of textual vignettes [52,53]. Bahmani [49] argues that vignettes enable “the researcher to collect data that is not accessible through other sources”, which supports our decision to use this method. Hazel [54] further reinforces this approach, emphasizing that vignettes allow researchers to present “actual cases of people and their behaviours, enabling participants to express their statements or viewpoints”. This approach aligns with the relational framework we aim to promote and with a crucial aspect of the HWL: empowering care workers (now co-designers) to reclaim their agency in both dialogue and action. This, in turn, enriches the research with meaningful, realistic change and highlights “important points from stories about those actors’ perceptions, beliefs, and attitudes” [54], which would not be easy to obtain if not from the perspective of those actors.

#### 4.1.1. Vignette 1: Enabling Learning and Personal Development (Social Value)

“*We can all learn from each other [and] create a culture where everyone is heard*”.(P1@CH#1)

Located north of Edinburgh, CH#1 features a beautifully designed and thoughtfully arranged layout, creating a calming, welcoming, and secure environment. The care home is committed to fostering inclusivity and diversity, valuing the unique traits and personalities of each staff member. This emphasis on embracing diverse backgrounds has earned the care home numerous awards over the years. The management takes great pride in the high quality of care and services provided by its skilled and passionate team. Potential residents and their families can have peace of mind knowing that their loved ones will be cared for by dedicated professionals. The institution’s commitment to creating a warm, welcoming environment, along with its outstanding track record and innovative approach to continuous professional development and staff well-being programs, sets it apart as a leader in the field.

In November 2022, a new training unit was inaugurated to enhance strategic resources and enable better conditions for staff to be fully trained in all aspects of care. The care home prioritises staff training through significant investment in learning and different aspects of relationality, supported by a structured plan for staff progression. This commitment has resulted in high staff retention rates and an overall satisfied workforce. Their distinctive approach to individualized, holistic care planning—considering both residents’ and care workers’ needs and preferences—is one of the many reasons this care home stands out. This provides strong evidence that a significant part of the care home business model is built on fostering a proactive and responsible approach, modelling and encouraging staff to adopt a collaborative culture. In this environment, they embrace their responsibilities with autonomy and flexibility, learning “from each other [and therefore] creating a culture where everyone is heard” (P#1@CH#1).

The staff mentioned that they receive regular and appropriate training, which makes them feel confident and well-equipped to handle any situation. They emphasized the importance of communication, noting that they “talk to each other to stay on the same page” (P#2@CH#1). This approach is crucial because it allows everyone to “learn from each other, shift their mindsets, and show respect for others” (P#2@CH#1). This appears to be contributing positively to reducing turnover and maintaining adequate staffing levels, ensuring that no one feels overwhelmed or overworked. This sense of fulfilment and the absence of excessive pressure foster a happy, patient, and compassionate team of staff who are deeply committed to their vital roles. The care home administration emphasizes the importance of communication, encouraging the proactive sharing of opinions and suggestions. Effective dialogue allows “others a chance to view or give an opinion to change” (P#3@CH#1), as noted by one team member.

This participatory process contributes to maintaining a high standard of care and a positive work experience, which is largely attributed to the open “communication with experienced staff” (P#3@CH#1). The manager hopes that the younger generation, guided by the experience and lessons of more senior employees, will uphold and possibly even elevate the high standards of care. This approach not only provides security and dignity but also positively impacts staff morale, well-being, and cohesion. Furthermore, those receiving care feel well looked after, valued, and respected. In this way, care transcends individual responsibility; it should be recognized as a shared societal obligation that contributes to social growth, cohesion, and overall social value.

#### 4.1.2. Vignette 2: Implementing a Culture of Openness and Support (Cultural Value)

“*It is important to look after your own health and well-being so that positivity can be passed on to others around you; you need to look after yourself to give good care to others*”.(P#5@CH#4)

CH#4 is in a village within the historic county. It is known for fostering a positive and supportive culture among its staff. The management team has been proactive in implementing meaningful changes to promote a culture of well-being within the care home. Notably, they introduced a Time Out/Bereavement room and organised yoga and meditation sessions, among other initiatives, as part of their efforts to innovate and enhance the well-being of both staff and residents. However, these efforts to promote change did not yield the expected results, highlighting the need for further reflection and strategic planning. For instance, one participant emphasized the importance of addressing basic needs before attempting to satisfy higher-level needs such as self-esteem. They noted, “this is important because if I don’t feel right physically, I won’t be able to perform at my best” (P#5@CH#4).

These elements were crucial pieces of the larger puzzle that enabled us to collaboratively co-produce a design pathway aimed at fostering a more supportive culture in this care home. One key activity we implemented was the Circle of Care, which encourages staff to adopt a holistic approach to care. The goal is to help them recognize the importance of self-care, emphasizing that “it’s essential to look after your own health and well-being so that you can pass on positivity to those around you; you need to care for yourself in order to provide quality care to others” (P#5@CH#4). In the end, five co-design sessions were conducted, encompassing the Performative Narrative Interview (PNI), Value Creation Process (VCP), Circle of Care (CoC), Deliberation, and Futuring through Newspaper Headlines of the Future (NHF). Following these sessions, and with management’s support, it was agreed to co-produce an action plan that highlights the creation of a new job role: the Staff Activity Planner.

The action plan aimed to promote a change in the care home’s ambiance. The team utilized air-purifying scent diffusers and Bluetooth-enabled speakers to create a calm, soothing, and uplifting environment, playing seasonal music via popular streaming apps. Additionally, the care home was redecorated with colours and scents reflecting the four seasons and festivities, enhancing reminiscence for residents and boosting morale for care workers. The most ambitious aspect of this plan was the co-production and implementation of an internal Relational Communication Training course. This course aimed to foster a more holistic, efficient, and effective communication style within the care home, enhancing both productivity and job satisfaction. The Relational Communication training focused on effective listening, constructive feedback, assertiveness, conflict resolution, and negotiation skills. The course materials were designed for easy transfer to an online learning management system, ensuring continuous access for staff and reinforcing their learning process.

#### 4.1.3. Vignette 3: Scaling-Up New Roles Through Co-Design (Economic Value)

“*I’m a co-designer now!*”.(P#1@CH#5)

Our third case reports on CH#5, a care home situated in a large plot of land. This home provides care for adults with dementia and dementia-related illnesses. It also supports adults with learning disabilities and behaviours which can present challenging situations between staff and residents. We started the co-design process in the care home by highlighting the excellent work produced with the poster probes—the initial design material introduced in the care home context. We emphasized that the two activities planned for the second session were based on this poster engagement. The first author served as the primary facilitator, while the second author acted as a secondary facilitator, observing and contributing when necessary. The second session included the Performative Narrative Interview (PNI) and the Value Creation Process (VPC). Our strategy involved guiding the co-designers through the activities while explaining the principles of co-design. This approach proved successful, as evidenced by one participant exclaiming, “I’m a co-designer now!” (P#1@CH#5). We conducted four additional sessions featuring Card Sorting (CS), a Sentence Builder Activity (SBA), the Circle of Care (CoC), Deliberation (DLB), and Building the Care Home of the Future (BCHoF).

One of the key challenges identified in this care home was a sort of intergenerational tension. A 52-year-old worker emphasized that “the older generation is the backbone” of the facility. This issue is particularly relevant as it has also surfaced in other research sites. However, it appears contradictory, given that younger care workers often assist the older generation in improving their digital skills. Some participants admitted to being technophobic, lacking even basic tools like email addresses. For example, P#3 mentioned that he tries to complete the online training on his own at home but often relies on his son or wife, while at the care home he relies on other staff, normally younger workers, to assist him with the technology due to “occasional glitches” (P#3 @ CH#5). He emphasised that in his role he could not progress to be a specialist without the training he had completed over time, mentioning that “it is regular, comprising mainly an e-learning approach” (P#3@CH#5), which he also does online at home. He achieved the Scottish Vocational Qualifications (SVQs), which are work-based qualifications to guarantee that he can perform his job to the national standards for the sector.

This insight highlights the potential to minimize intergenerational tension by fostering the exchange of skills and experiences between generations, recognizing such types of support in the sociotechnical complexity of ‘solutions’. Leaders in care settings can leverage this by adopting strategies that develop critical digital capabilities while bridging generational gaps. For instance, they can promote experiential learning, where staff members gain knowledge through hands-on experience and reflection together, rather than individually. Leaders can lean into this by fostering a culture of openness and continuous improvement, encouraging staff to share ideas, experiences, and mistakes.

The outcomes showed that staff felt more invested in their responsibilities after participating in co-producing the solutions they would implement. For example, the co-design activities in this care home revealed that, beyond merely providing care workers with tools and training, involving them in collaborative problem-solving and decision-making (i.e., co-producing change) builds their confidence, deepens their understanding of the context, and strengthens their commitment to their roles. As a result, they better comprehend their daily tasks and can handle crises with greater flexibility. This approach was particularly effective in achieving a more balanced workload distribution. For instance, optimizing processes saved time, which could then be used to deliver higher-quality care to residents and give care workers more time to rest. Furthermore, the co-design activities encouraged staff to think beyond their specific roles, demonstrating that co-design had a significant impact on the care home’s business strategy, reshaping its economic value.

## 5. Discussion

Drawing on our empirical material and using the Design Research Value Model proposed by [45], we will next illustrate how the co-production of change can be achieved through co-design, addressing unmet complex needs triggered by ‘wicked problems’ which tend not to be best addressed with the conceptualisation of potential responses as simple. We have demonstrated that these needs are more likely to be effectively addressed when the design response is context-specific. In this article, we define such responses as ‘wicked solutions’, which are co-produced by different actors interacting in a more relational perspective. These are bespoke design responses which tend to empower actors—care workers, in this case—to regain their agency to speak out and act within their social environment [19]. The RF enabled co-designers’ novel opportunities to address what Rogers stressed to be the need to “identify and articulate the significant roles that design research plays in generating social, cultural, economic and environmental value” [45]. Figure 9 presents an infographic summarising the three forms of value. First, we define these values based on Rodgers et al., followed by an outline of the necessary requirements for fostering a more relational participatory process. This includes a focus on the tangible and intangible co-production requirements, as well as the expected direct and indirect outcomes of these interactions. Finally, we provide a more detailed account of the three values that have emerged through this research.

### 5.1. Co-Designing Social Value for Care

Social value co-creation requires significant investments in time, energy, and resources from all involved parties. Those participating must be open to sharing ideas, debating their merit, and ultimately negotiating a shared understanding and direction. This is no easy task, as it involves overcoming obstacles such as pride, ego, and control and commitment to care for residents. However, the benefits of social value co-production can far outweigh these challenges. This collaborative design process encourages participants to think innovatively and consider fresh perspectives and ideas, and helps foster a sense of community and shared identity. As participants engaged in these processes, they were often challenged to develop and refine their skills and abilities. They learned to communicate effectively, negotiate, and compromise, all of which are valuable social skills, contributing to individual personal growth and development, while enabling social value co-production through “face-to-face participation, real-time interaction, and alignment towards a common goal” [45].

### 5.2. Co-Designing Cultural Value for Care

Within the scope of this article, we refer to cultural value as “the tacit knowledge embodied in social processes” [45], which can take a variety of forms, including localised norms, languages, and practices, all contributing to a sense of purpose and belonging. As such, a shared organisational culture is fundamental to help deliver high standards of care because it contributes to training care workers in a particular context and guiding them in their interactions with others, serving as a source of inspiration and creativity, as well as allowing them to foster critical thinking. Evidence produced by this research helped us demonstrate that those individuals who interacted with their colleagues through co-design activities learned to interpret their organisational context, and gained deeper understanding of concealed nuances and complexities. This in turn supported their phronetic (situated) wisdom, or the discernment to know the best course of action in any given circumstance. This can be aligned with Rodgers’ perspective that “cultural engagement contributes to a greater shaping of reflective individuals” [45].

### 5.3. Co-Designing Economic Value for Care

According to Rodgers et al., “[d]esign research can also create cultural value through participatory learning experiences that enhance individuals’ abilities to gain skills, knowledge and awareness” [45]. The vignettes we used to illustrate some of our results show that design research can generate economic value. Design-centric organisations are adopted to support empathy and reflexive understanding between users in a shared situation, which helps them understand their stakeholders’ experiences and incorporate the insights into their product or service development process (human-centred approach). Evidence produced by this research shows that through co-design activities, it is possible to pave the way to improved retention and recruitment, while minimizing effects of the care crisis. Here we argue that that economic value can be achieved through improved relationships in ‘circles of care’, and the recognition of such relationships in the framing of design responses for this sector. The key point to underline is that services, products, and technologies [55] framed around individual users miss the relational point, while those designed for assemblages of support are more likely to be adopted and adapted.

## 6. Future Work

In the UK, the care sector is primarily made up of independent (and often small) businesses of varying size, organisational structure, culture, and context. All are rightly subject to strict legislation relating to care and safeguarding, and the majority are dependent on incomes per resident that are set by national and local governments. This limits business flexibility and complicates what is already a resource-poor environment (in finances, time, and workforce). As we look toward the sector’s future, it is important to consider that each care home is itself a ‘complex system’ and has its own contextually derived concerns. To “perform research into complex systems in which power law distributions are in operation there is a need to interpret the processes of dynamicity” [35]. This, we suggest, requires us to act more relationally in the field, in which those impacted by such forces should be placed at the core of the research.

## 7. Conclusions

We contribute an original framework which enables the co-production of social innovation and social change, namely for the care sector and, potentially, other complex settings. The Ripple Framework has assisted individuals (care workers) and industry (care homes) with the opportunity and flexibility to co-produce social, cultural, and economic value within the care sector, while supporting design research to expand its scope of action when facilitating tools to reframe relevant assumptions, such as the stigma that affects the care sector. This is an important conceptual and applied approach in driving innovation and growth in the healthcare industry. Reach can be significantly improved through ‘radical participatory design’, which allows for better understanding of actors in context (that is, a relational approach). In this article, we have illustrated how, when assisted to reinterpret their own tasks, these actors can significantly contribute to co-produce bespoke design responses to problems they have themselves identified. We conclude by stressing six fundamental steps in any relational co-design process to co-produce care sector changes:Providing care workers with the necessary resources and autonomy to act.Encouraging collaboration: teamwork and cooperation between care workers can lead to the exchange of ideas and perspectives, fostering creative solutions.Encouraging communication: open and constant communication across the care home structure, from management to workers. This will ensure everyone is on the same page and working towards a common goal. We may refer to this as the flattening of the hierarchical structure.Supporting innovation: providing a supportive environment that encourages risk-taking and out-of-the-box thinking is essential to fuel innovation and change.Ensuring inclusivity: bringing in diverse perspectives can lead to more well-rounded and unique responses.Delegate responsibilities: each member should be given specific roles to highlight their strengths and ensure equal involvement in co-producing change.

## Figures and Tables

**Figure 7 ijerph-21-01521-f007:**
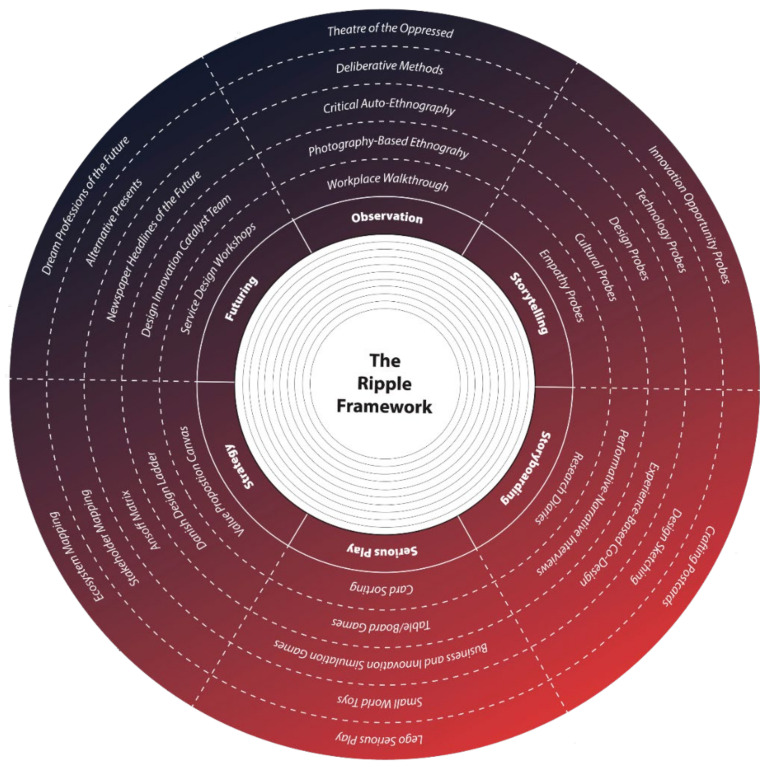
The current interactive version of the Ripple Framework) [25].

**Figure 8 ijerph-21-01521-f008:**
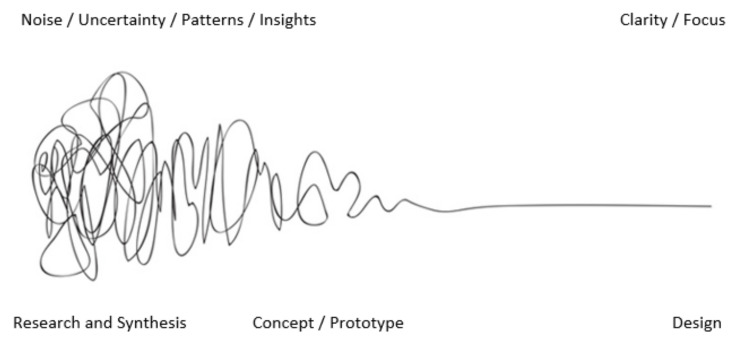
The Design Squiggle: As originally proposed by Damien Newman [34].

**Figure 9 ijerph-21-01521-f009:**
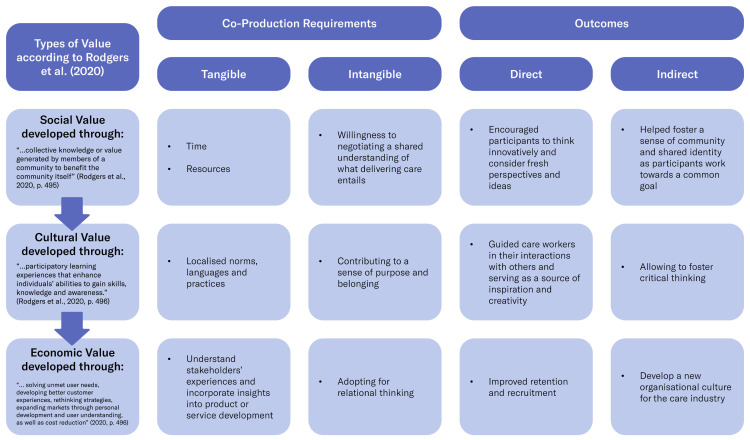
Adapting the Value Model proposed by Rodgers et al. [45] to HWL.

**Table 1 ijerph-21-01521-t001:** Co-design stages.

Stage	Intended Action	Approach/Activity	Intended Outcomes
1	Observational	Walkaround	Know the actor and the context
2	Teaser/Grabber	Poster Probe	Illustrate and Exchange
3	Deepening	Performative Narrative Interview	Engage and Unlock direction and expectations
4	Interactive/Doing	Deliberation	Refinement
5	Changing	Postcards from the Future and Postcards to the Powerful	Alternative Future

## Data Availability

HWL data is available at the Kings College of London data repository.

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
