# Peer review of "The Ripple Framework: Co-Producing Social, Cultural, and Economic Value in Care Through a Generative and Relational Approach"

_ijerph, 2024, doi:10.3390/ijerph21111521_

Round 1
Reviewer 1 Report
Comments and Suggestions for Authors
I wanted to like this project that puts staff at the centre of the research and employs an interesting variety of methods to do so. However, I found the sometimes confusing language and lack of information on participants and context frustrating.
Until I was halfway through, I could not figure out whether the issue was the codesign of the project or the care work itself. Are they testing and retesting the design of the research or of the care work approach (@169). What are the power relationships at issue? What does the “true value of care (@263) mean? Who are the staff in these workplaces and why include only those staff aged 50 and over? What is their place in the workplace hierarchy? What about gender, race, ethnicity? We are told once about a man but little else in relation to staff composition. Who are the residents? We are told one home (Vignette 3) has residents with dementia but what about the others? Indeed, we learn little about the workplaces in terms of things like the class or capacities of the residents. What does it mean to say staffing levels “appear adequate” (@ 315)? What about job security and precarity, which other literature would suggest has an impact on the ability to participate in such innovative approaches.
In short, I think the paper needs greater clarity about purpose and outcome and much more detail on the context and participants.
Author Response
Dear Reviewer 2,
Thank you for your thoughtful comments and valuable suggestions for improving the paper.
I fully agree with all your comments/suggestions; they were pertinent and significantly contributed to enhancing the quality of the manuscript. I am pleased to inform you that your suggestions have been carefully incorporated into the revised draft. However, there is one comment that we decided to keep things as per the original manuscript, i.e.:
Comment: " Indeed, we learn little about the workplaces regarding things like the class or capacities of the residents."
Response: This aspect was not covered, considering that the study's scope did not include the residents. The focus was to work with the care workers to understand the sector from their perspective. Aside from this case, all the other comments/suggestions have been addressed.
Thank you
Obs. all the other comments have been addressed directly in the manuscript
Reviewer 2 Report
Comments and Suggestions for Authors
1. The introduction lacks a sufficient review of recent literature. To strengthen the foundation of the problem statement and justify the research focus, it would be beneficial to include a review of 10-15 relevant studies from the past five years.
2. The authors stated, "...with the 31 participants who engaged with us over the ten months of co-design." It would be helpful to clarify how the study's target group was identified and provide more details on the selection criteria and rationale.
3. Presenting the study's contributions as an infographic would enhance clarity and make the findings more accessible to readers. Consider visually summarizing key insights and outcomes to improve engagement.
Author Response
Dear Reviewer 2,
Thank you for your thoughtful comments and valuable suggestions for improving the paper. I fully agree that your insights were pertinent and significantly contributed to enhancing the quality of the manuscript. I am pleased to inform you that your suggestions have been carefully incorporated into the revised draft.